# Zebrafish Tools for Deciphering Habenular Network-Linked Mental Disorders

**DOI:** 10.3390/biom11020324

**Published:** 2021-02-20

**Authors:** Anja Bühler, Matthias Carl

**Affiliations:** Department of Cellular, Computational and Integrative Biology (CIBIO), University of Trento, 38123 Trento, Italy

**Keywords:** zebrafish, habenula, behavior, mental disorder, asymmetry, neurogenesis, Wnt, compound, pharmacology

## Abstract

**Simple Summary:**

Everything that we think, feel or do depends on the function of neural networks in the brain. These are highly complex structures made of cells (neurons) and their interconnections (axons), which develop dependent on precisely coordinated interactions of genes. Any gene mutation can result in unwanted alterations in neural network formation and concomitant brain disorders. The habenula neural network is one of these important circuits, which has been linked to autism, schizophrenia, depression and bipolar disorder. Studies using the zebrafish have uncovered genes involved in the development of this network. Intriguingly, some of these genes have also been identified as risk genes of human brain disorders highlighting the power of this animal model to link risk genes and the affected network to human disease. But can we use the advantages of this model to identify new targets and compounds with ameliorating effects on brain dysfunction? In this review, we summarise the current knowledge on techniques to manipulate the habenula neural network to study the consequences on behavior. Moreover, we give an overview of existing behavioral test to mimic aspects of mental disorders and critically discuss the applicability of the zebrafish model in this field of research.

**Abstract:**

The prevalence of patients suffering from mental disorders is substantially increasing in recent years and represents a major burden to society. The underlying causes and neuronal circuits affected are complex and difficult to unravel. Frequent disorders such as depression, schizophrenia, autism, and bipolar disorder share links to the habenular neural circuit. This conserved neurotransmitter system relays cognitive information between different brain areas steering behaviors ranging from fear and anxiety to reward, sleep, and social behaviors. Advances in the field using the zebrafish model organism have uncovered major genetic mechanisms underlying the formation of the habenular neural circuit. Some of the identified genes involved in regulating Wnt/beta-catenin signaling have previously been suggested as risk genes of human mental disorders. Hence, these studies on habenular genetics contribute to a better understanding of brain diseases. We are here summarizing how the gained knowledge on the mechanisms underlying habenular neural circuit development can be used to introduce defined manipulations into the system to study the functional behavioral consequences. We further give an overview of existing behavior assays to address phenotypes related to mental disorders and critically discuss the power but also the limits of the zebrafish model for identifying suitable targets to develop therapies.

## 1. Introduction

The bilaterally formed neuron clusters of the habenula in the vertebrate dorsal diencephalon are in the center of a wide-spanning neural circuit which coordinates information processing between fore-, mid- and hindbrain nuclei [1,2,3,4,5]. This is achieved by a number of neurotransmitters [6]. Acetylcholine, substance P, glutamate, dopamine, and serotonin levels are influenced by habenular neuron activities and hence crucial processes such as motivation, motor, cognitive and related abilities rely on the functionality of this neural network. Although the habenular circuitry has been subject to investigations since the beginning of the 20th century, only recently its implications in a growing number of psychiatric disorders have come into focus. These range from substance abuse, anxiety, and mood disorders to pathophysiological syndromes like autism, bipolar disorder, and schizophrenia [4,7,8,9] (also reviewed in [5,10]). One prominent example is habenular hyperactivity in humans with major depressive disorder, for which several disease model systems are available [11,12,13,14]. The mammalian habenula can largely be subdivided into medial (MHb) and lateral habenular nuclei (LHb). Deep brain stimulations (DBS) to inactivate the LHb increases monoamine levels and result in either full remission or at least alleviation of some major symptoms of otherwise treatment-resistant patients [15,16,17]. Furthermore, studies in rats indicate that DBS of the LHb effectively reduce cocaine and sucrose self-administration in a rodent model of substance abuse [18,19]. Intriguingly, non-invasive treatment with the antidepressant drug ketamine was recently uncovered to ameliorate the condition in rodent models of depression by inhibition of burst firing of LHb neurons [8,9]. This progress highlights the great potential of the habenular neural circuit as a therapeutic target.

In addition to these advances, recent findings regarding the genetic basis of habenular neurogenesis and axonal targeting have opened up exciting links between risk genes, the habenula, and mental disorders. In particular, studies using the zebrafish animal model system have revealed that the risk genes for schizophrenia and autism, tcf7l2 and wif1 [20,21,22,23], are crucial regulators of habenular neurogenesis and axonal targeting [24,25]. Some of these works have now begun to be repeated in the mouse model [26] highlighting the evolutionary conservation of important aspects of the mechanisms underlying habenular neural circuit formation. Even though causative genes of mental disorders normally differ, and syndromes are defined by distinct behavioral abnormalities, patients also display striking similarities such as anxiety, disturbed social behavior, reduced motion, and mood dysfunction [27,28]. Many of these behaviors can be studied in animal models including the zebrafish. Even if the full complexity of human behaviors cannot be mimicked in animals, the modeling of some typical aspects might serve as an excellent readout for identifying disorder related risk genes.

Here we critically summarize the potential of the zebrafish model to pave the path for developing new approaches to target mental disorders related to the habenular neural system with a particular focus on available behavior test systems.

## 2. The Zebrafish Habenular Network and Tools for Manipulation

### 2.1. The Zebrafish Habenular Circuit

To identify suitable targets for ameliorating habenular network-related genetic disorders, the molecular and genetic mechanisms underlying its function need to be unraveled. The ease of genetic manipulations, the transparency of the rapidly developing numerous offspring, and their rather complex behaviors make the zebrafish particularly suitable for this endeavor. Furthermore, all major neurotransmitters, their corresponding receptors, and transporters as well as enzyme synthesis and metabolic pathways are conserved between mammals and teleosts [29] highlighting the translational potential of discoveries made in this model system. The bilateral habenulae are each subdivided into two main nuclei named medial and lateral habenula in mammals, which correspond to the dorsal and ventral habenula in teleosts, respectively based on marker gene expression, tracing experiments, and axonal connectivity [30]. The ventral habenulae (vHb) consist of a single neuronal population which only forms in the presence of functional Tcf7l2 [31]. The dorsal habenulae (dHb) of zebrafish are made up of two neuronal subpopulations, the lateral (dHbl) and medial (dHbm) habenular subnuclei (Figure 1). Several signaling pathways have been shown to influence the neuronal diversity of the habenula. Initially, Wnt/beta-catenin signaling is involved in the proliferation of multipotent dHb precursor cells left and right adjacent to the medially positioned pineal organ [32]. A recent study showed that subsequently Wnt signaling is required to be largely inactive to prevent the delay of neuronal differentiation and the resulting improper dHb neuron generation in favor of dHbm neurons [25]. Habenular precursor cells are protected from the premature initiation of Wnt/beta-catenin signaling by the secreted Wnt inhibitory factor Wif1, which binds Wnt ligands [33]. Intriguingly, Wif1 acts with Wnt signaling in a regulatory feedback loop [25]. This discovered feedback mechanism, which had previously been postulated for cancer cells [34], is thought to allow the system to quickly react to the dynamic changes in the environment of the habenular precursors. Indeed, the mid-diencephalic organizer (MDO) anteriorly adjacent to the habenular precursor pools is a prominent source of secreted Wnt ligands at this stage of embryonic development [35]. During this time of Wnt inhibition, the precursor cells are influenced by Notch signaling and Notch inactivation in *mindbomb/E3 ubiquitin ligase* mutants causes premature differentiation of habenular precursors cells into dHbl neurons [36]. Nodal signaling initially supports the generation of a few dHbl neurons, which in zebrafish and many other vertebrates occurs only in the left brain hemisphere [37]. Nodal also biases some cells of the pineal to migrate from the center of the dorsal diencephalon in between the developing habenulae towards the left side influenced by FGF signaling [37,38,39,40]. These parapineal cells appear to secrete yet unknown molecules to promote dHbl neuron generation, which requires functional sox1a [41,42,43,44]. Based on epistasis experiments, one function of these proteins appears to be the suppression of Wnt/beta-catenin signaling in the left-brain hemisphere at a time when habenular precursors become post-mitotic and the expression of *wif1* begins to decrease [24,25]. Hence the function of Wif1 to prevent premature Wnt signaling activation appears to be handed over to the action of the parapineal derived molecules, at least on the left side of the brain. Suppressed Wnt/beta-catenin signaling on this side results subsequently in dHbl neuron generation, while active Wnt signaling promotes dHbm neuron development on the right. Therefore, the default fate of a dHb precursor cell in a Wnt signaling devoid environment is to become a dHbl neuron. Recent evidence from mouse experiments suggests that this function of Wnt/beta-catenin signaling to influence habenular neuron diversity in postmitotic precursor cells is evolutionarily well conserved [26]. A major difference between these two animal models is that in zebrafish, Wnt/beta-catenin is suppressed in the environment of left-sided nascent habenular neurons in the presence of the parapineal, which results in clear differences in the size of neuronal populations between the left and right side. Mammals do not develop parapineal cells and generate rather subtle left-right asymmetry of the habenulae [45,46]. In contrast, functional asymmetries in the network are present, but the mechanisms underlying the pronounced functional lateralization as well as its importance remain largely unknown [47]. These aspects are being explored in zebrafish [4,48,49]. For example, the two dHb neuron populations show distinct stimulus-dependent lateral bias in activation, indicating a differential role in modulating behavior. Dreosti and colleagues demonstrated that most dHbl neurons respond to visual stimuli, whereas predominantly dHbm neurons respond to olfactory stimuli [48]. This bias is lost if the fish develop symmetric habenula.

The current picture of dividing the dorsal habenula into two neuronal populations is certainly over-simplified and the neuronal diversity in this brain structure is much more complex as evidenced recently by single-cell RNA-seq analysis [50]. Some genes are expressed exclusively in one particular neuron population, while several habenular genes are transcribed partially overlapping in both or even all habenular neuron clusters, including ventral habenular neurons [31,51]. The existence of habenular neurons within each subnucleus expressing different sets of genes may suggest functional diversity within the clusters or perhaps dynamic flexibility in case of loss of neurons. In addition, particular gene combinations may also influence the pathfinding and/or precise targeting of habenular efferent axons within the interpeduncular nucleus (IPN) about 300 µm away from the dorsal diencephalon in the ventral midbrain [52] (Figure 1). Here the axon terminals segregate laterotopically within the IPN dependent on their subnuclear origin [2,51,53]. Axons from dHbl neurons mainly target the dorsal IPN, whereas axons from dHbm neurons innervate the ventral IPN. Unlike in mammals, in which the IPN has been well characterized [54], revealing the precise IPN subdivisions and the connectivity of all habenular axons within their IPN targets has only been started in zebrafish [2,55,56]. This is in part due to the lack of suitable genes and transgenes to label IPN cells and cell populations. Moreover, the positioning of the nucleus deep down in the midbrain hampers easy microscopic accessibility for live imaging. Habenular efferent axon pathfinding depends on the axon guidance receptor nrp1 [57], which in turn is regulated by a recently uncovered late function of the parapineal organ [44]. Additionally, thalamic neurons posterior to the developing habenulae, which are the source of ventral habenular neurons [31], are important for axonal network formation [57]. In their absence, the initiation of habenular axon extension is delayed and their left-right synchronized extension behavior is disrupted. On the way to their midbrain target nucleus, the extending axons communicate across hemispheres using a second, earlier developing neural network originating in the thalamus [52]. Unilateral disruption of this network results in the loss of IPN innervation by all habenular axons. The molecules involved remain to be uncovered.

### 2.2. Zebrafish Habenular Network Manipulations

Although much remains to be learned about habenular neural network formation, our knowledge gained over the past two decades left us with various tools to manipulate the network in zebrafish and study the functional consequences (Figure 1). Changing the laterality of habenular asymmetries is rather trivial by manipulations of the nodal signaling cascade using genetics or pharmacology. Another method is to simply raise embryos at low temperature for some hours around the beginning of gastrulation (cold shock treatments), which presumably also influences nodal signaling even if indirectly (Figure 1) [58,59,60,61]. However, these techniques cause laterality effects in both body and brain. Specific changes to habenular laterality are difficult to induce, although recent parapineal cell transplantations nourish hopes [44]. Interference of dHb-IPN information flow has been successfully carried out. In elegant experiments, the Okamoto lab has applied the transgenic tetanus neurotoxin (TeNT) technique to selectively silence neurotransmission of dHb subnuclei and analyze zebrafish behavior (Figure 1) [62,63]. Various methods and resources are available to cause symmetric dorsal habenular neuron populations to develop such that one neuron population is predominantly generated on both sides of the brain. The perhaps most frequently applied technique is the focal laser ablation of the parapineal cells, which causes more dHbm neurons to be generated on the left-brain hemisphere at the expense of dHbl neurons (double right symmetric habenular phenotype) (Figure 1) [2,41,42,51]. A recent discovery highlights the importance of timely precision of such ablations as a delay would only influence habenular axon targeting rather than neurogenesis [44]. This finding may in turn open new paths when investigating the molecular mechanisms underlying habenular axon targeting. Genetically, interference with sox1a or tbx2b causes double right symmetric habenular phenotypes due to the loss of parapineal cell function to influence habenular neurogenesis or their development respectively (Figure 1) [44,64]. In summary, a number of genetic as well as physical techniques are available to cause symmetric habenula with increased dHbm neurons, which appear largely specific and allow not only testing of zebrafish larva but also growing up manipulated embryos to adulthood and analyze their behavior. In contrast, tools for manipulating the habenula to cause a double left habenula phenotype (development of more dHbl neurons on the right at the expense of dHbm neurons) are rather limited to date (Figure 1). Genetic mutants for Notch or Wnt signaling, which develop this kind of habenular symmetry [24,36,58] are embryonic lethal, although mutants for *tcf7l2* at times die only after around six weeks [65]. However, *tcf7l2* mutants also lack the vHb, which needs to be taken into consideration for functional studies [31] The temporal control of Wnt/beta-catenin signaling and its involvement in habenular neuron specification has revealed a narrow developmental time-window in which the blockage of Wnt signaling within 1–2 h using the IWR compound causes an increase of dHbl neurons on the right side (Figure 1) [24]. Such compound treatments might provide a valuable tool for transiently regulating signaling pathways to cause largely specific alterations in neuronal differentiation. In summary, recently gained knowledge on habenular neural network development allows researchers to introduce defined alterations into the neuronal composition and their axonal targeting. As a next step, it is important to study the impact on fish behaviors and whether they mimic certain aspects of those observed for mental disorders.

## 3. Behavioral Tests in Zebrafish to Study Human Habenula Linked Disorders

Rodents are mainly chosen when it comes to model behaviors related to human psychiatric disorders. However, also zebrafish exhibit a wide range of behaviors, behavioral responses to stimuli, and well-developed cognitive and complex decision-making abilities. Similar to rodent models, zebrafish furthermore show a high sensitivity to pharmacological modulation of behavioral responses [66,67,68,69]. Behavioral tests to investigate a broad range of human psychiatric disorders, neurodevelopmental disorders, and neurodegenerative diseases have been generated. They can be readily combined with sophisticated automated analysis and high-resolution video-tracking systems [70,71]. Many behaviors including anxiety, fear, and stimuli dependent learning can be assessed as early as in free-swimming larval stages, whereas social behavior like shoaling [72], conspecific directed aggression [73], mating [74], and conspecific preference [75] develop with age. Here we focus on existing behavioral tests with relevance for human habenula linked mental disorders such as anxiety, depression, and social interaction (Table 1). It is important to emphasize at this point that zebrafish behavior is not a turnkey system and obtaining consistent results requires meticulous preparation, profound knowledge, and time for setting up a robust test system (see also the concluding Section 4).

### 3.1. Behavioral Tests for Anxiety

Anxiety disorders including generalized anxiety disorder, panic disorder, phobia, social anxiety disorder, and separation anxiety disorder are complex psychiatric conditions determined by multiple genetic and environmental factors [110]. They represent one of the most prevalent groups of psychiatric disorders and patients suffer from severe impairment of life quality [111]. The habenula is a processing center for negative reward information and has been demonstrated to regulate aversive and anxious states in animal models (reviewed in [4,5,10]). Behavioral anxiety paradigms are therefore interesting tools to study habenula function in human mental disorders.

Analogous to behavioral tests in mammals, a number of tests for zebrafish anxiety have been established (Table 1). The maybe best known and most used test to assess zebrafish anxious behavior is the novel tank test, which is, like the related zebrafish open field test (below), comparable to the open field test or the elevated plus maze in rodents [76,77,78,79,80]. This test assesses the innate diving and explorative behavior of adult zebrafish placed in a novel tank. After a short phase of bottom-dwelling (geotaxis), in which the fish acclimates to the new environment, exploration by swimming in the upper parts of the new tank starts. Decreased explorative behavior (latency to reach, time spent, and number of entries into the upper part of the tank) as well as erratic swimming and increased freezing bouts are measures for anxious behavior typically analyzed. Besides the novel tank test, the open field test is commonly used to investigate anxiety-related behaviors (Table 1) [81]. The open-field test differs slightly from the novel tank test such that instead of observing the “vertical” behavior (like diving) from the side, the “horizontal” movement is analyzed from above. Time spent in the outer zones of the tank or “wall-hugging” behavior (thigmotaxis) and hyperactive swimming are characteristics for anxiety in this test [81]. The behaviors described in both tests are highly sensitive to anxiolytic and anxiogenic drugs, with anxiogenic drugs decreasing exploration and increasing erratic swimming, thigmotaxis, and freezing (Table 2). Conversely, anxiolytic drugs reduce these anxious behaviors and increase normal exploration behavior (Table 2, reviewed in [112]). Another behavioral test for novelty-based anxiety is the light-dark box (dark/light test in rodent models) (Table 1). It takes advantage of the natural tendency of adult zebrafish to avoid bright areas and their preference to spend time in the dark areas (scototaxis), when placed in a tank with a bright and a dark compartment [113]. Avoidance of the bright area is measured by the time spent in the area and the number of crosses between the bright and the dark compartment. When confined to the bright area, animals show robust signs of anxiety such as increased burst swimming, thigmotaxis, and freezing. Anxiogenic drugs increase the dark preference as well as thigmotaxis, erratic swimming, and/or freezing in the white compartment, whereas anxiolytic drugs reduce some or all of these behaviors [89,113,114]. The preference of adult fish for the dark is in line with observations from the rodent model in the light/dark test [115]. But the situation is complicated by reported occasional incidences of unexplained light preference in adult zebrafish [81]. Persisting light preference is the natural behavior for larval fish which display active phototaxis toward light [91,92]. Furthermore, Schnörr and colleagues demonstrated that sudden light-to-darkness transitions induce scototaxis in larval zebrafish. As in adults, the anxious response is highly sensitive to anxiolytic drugs and can be significantly reduced by treatment with diazepam, buspirone, and ethanol (Table 2) [93,116,117].

Besides the novelty-based paradigms, paradigms based on negative stimuli associated with imminent danger such as the presence of a predator can be used to elicit fear and anxious behavior which follows the same principle of the predator (or predator image- and odor-) exposure test in rodents (Table 1). Different strategies of presenting the stimuli have been developed to better standardize the evoked responses under laboratory conditions. Moving images of a predator fish [85,118], robotic predator fish [84], a shadow representing a bird/bird silhouette or a dot increasing in size shown from the top (mimics a fast-approaching object from above) [69] or alarm substance [80,87] are being used. Anxiolytic drugs can counteract fear and anxiety-related behaviors (Table 2), although perhaps not all of the behaviors observed in the novelty-based paradigms are induced by aversive cues [69]. Another behavioral pattern induced by aversive cues is the startle response, which is evoked by an abrupt onset of a stimulus such as vibration, light, sound, or touch (Table 1). It is a characteristic escape behavior which is initiated by acceleration away from the stimulus by a “C-bend” of the body and is often followed by fast zig-zag swimming near the bottom of the tank in adults or fast turning and burst swimming in larvae [114]. Ethanol reduces the startle response in both adults and larvae (Table 2) [119,120] demonstrating sensitivity to anxiolytic drugs.

In the cued fear conditioning test, the association between an aversive stimulus e.g., mild electric shock (unconditioned stimulus, US) and a cue e.g., red light (conditioned stimulus, CS) is tested (Table 1). After repeated presentation of the stimuli together, fish quickly overcome an initial freezing response and display escape behavior in response to the CS. Intriguingly, dHbl-dIPN silenced adult zebrafish respond with increased freezing rather than with an escape response [55]. Moreover, genetic disruption of the dHb-IPN pathway in larval zebrafish leads to a deficit in active avoidance learning in cued fear-conditioning tasks in which the larvae had to swim away from the light (CS) in order to avoid the electric shock [95]. These results were initially interpreted as a tendency for passive instead of active coping in habenula lesioned fish [4]. Mathuru and colleagues however demonstrated elevated anxiety baseline levels (increased bottom-dwelling in novel tank test and disproportionate strong fear response when presented with multiple mildly stressful stimuli) in dHb silenced fish as a plausible cause for the increased freezing response to the CS [132]. Moreover, larvae with a disrupted dHbl-dIPN pathway do not display a normal startle response/escape response but also increased freezing when confronted with a single, non-conditioned aversive stimulus (electric shock) (Table 1). Interestingly, only disruption of the dHbl-dIPN connection but not the dHbm-vIPN pathway leads to anxious behavior and correspondingly mainly the dHbl neurons are activated post-shock [94].

One aspect of the startle response which might be of particular interest to test habenula linked disorders in the fish is the so-called prepulse inhibition (PPI) (Table 1). PPI influences the startle response by an attenuation of the startle when a weak non-startling stimulus is presented before the startling stimulus [133]. PPI is impaired in many human disorders including Tourette’s syndrome [134] and schizophrenia [135] and can be reversed by antipsychotic drugs in schizophrenic patients [136]. Intriguingly, antipsychotic drugs such as apomorhine, haloperidol as well as ketamine influence PPI in zebrafish larvae (Table 1) [100,101,137].

### 3.2. Behavioral Paradigms for Depression

Hyperactivity of the habenula has been linked to depression in human patients as well as several animal models of depression including the zebrafish [8,12,109]. Depression or major depressive disorder is a common and complex mood disorder caused by genetic and environmental factors [138,139]. Characteristic symptoms are sadness/low mood, fatigue, impaired cognitive functions and decision making, thoughts of suicide, anhedonia, and weight fluctuations [140].

Commonly applied tests in mammalian models are the behavioral despair test [141,142], the chronic unpredictable stress (CUS) [143], chronic mild unpredictable stress (CMUS) [144], and chronic social stress [145]. In the last decade, similar paradigms have been tried in zebrafish. When applying CUS paradigms, the fish are exposed to different stressors for 1 to 5 weeks (e.g., restraint stress, temperature stress, social isolation stress, predation stress, electric shock). Afterwards, the fish can be tested with a battery of behavioral tests. Increased anxiety levels, impaired cognitive function, and disturbed social interaction were demonstrated following CUS (Table 1) [105,106,107,108]. Intriguingly, the depression-like phenotypes can be investigated for physiological measures (e.g., elevated cortisol levels) and are sensitive to treatment with clinically relevant antidepressants such as fluoxetine, bromazepam, and nortriptyline, suggesting CUS as a suitable depression model in zebrafish (Table 2) [108,146].

Prolonged exposure to stressors and aversive situations also impacts how animals and humans adapt to stressful situations. When active coping mechanisms e.g., escaping the situation/the aversive stimulus repeatedly do not achieve to counteract the stressor, these actions will be suppressed in favor of passive coping mechanisms and can manifest as hopelessness or helplessness, a key symptom of depression. The switch between active and passive coping mechanisms can be investigated in behavioral challenge assays in which animals are exposed to an inescapable aversive environment [141,142]. A study by Andalman and colleagues in which larval zebrafish were exposed to inescapable shocks demonstrated an involvement of the ventral habenula (vHb) in passive coping behavior (Table 1) [109]. After initially trying to unsuccessfully escape the shocks, the fish entered a state in which they moved significantly less than unshocked fish, signaling the switch from active to passive coping. This passive state was correlated with progressive hyperactivation of the vHb in whole-brain scans. Intriguingly, vHb hyperactivation was prevented by prior treatment with the antidepressant drug ketamine.

In rodent social defeat paradigms, subordinate mice are exposed to daily bouts of social defeat by a more aggressive conspecific. Continuous social defeat leads to decreased social behavior, elevated anxiety level, and reduced sucrose preference [145,147]. A recent study used male fighting behavior as a social defeat paradigm in zebrafish. Losers were repeatedly exposed to aggressive attacks from winners and lost their motivation to fight and to win, even against “weaker” opponents (Table 1) [104]. Intriguingly, losers in this model showed elevated vHb activity, further indicating that vHb activity represents an aversive expectation value [104]. The results of the behavioral challenge and the social defeat paradigm suggest a pivotal role of the vHb in passive coping and are further substantiated by an earlier study by Amo and colleagues in which they demonstrated a modulation of the vHb-MR pathway in the adaptive fear behavior in response to dangerous environments [148]. Moreover, activation of the vHb in zebrafish depression models are consistent with hyperactivation of the habenula in humans and rodent depression models [8,149].

In summary, animal models can hardly recapitulate the full range of human emotions, but besides genetic and pharmacological models of depression, several behavioral paradigms for depression in the zebrafish are established and can be validated through administration of antidepressants (Table 2) and physiological measures (e.g., cortisol levels) [108].

### 3.3. Social Behavior Paradigm

A particularly attractive characteristic of the zebrafish is their conspicuous social behavior. Impairment of social behavior and interaction is a core symptom of mental disorders such as autism spectrum disorder (ASD) and schizophrenia. A recent study found that hyperactivation of the LHb leads to decreased sociability in rodents [7], strengthening the link between the habenula and complex human mental disorders.

Zebrafish display complex social behavior and therefore social engagement and spend most of their time in shoals. Shoals are mixed-gender groups with structured hierarchies which are influenced by their social experiences (winner/loser effect) [150,151] and preferences for kin and familiar conspecifics [152,153]. Shoaling behavior can be easily evoked and analyzed with 2 and 3D automated video tracking and several parameters such as nearest neighbor distance (NND), inter-individual distance (IID), swim speed, bottom dwelling, and shoal polarity can be reliably measured [96,154]. Driver factors like novelty, chronic stress [105,131], predator, or alarm pheromone exposure [87,155] evoke specific changes in shoaling including tightening of the shoal, increased thigmotaxis, and bottom dwelling (Table 1). Moreover, shoaling can be also measured in heterogeneous shoals in which a mutant/treated fish is placed in a wildtype/control shoal [156]. Notably, several drugs such as ethanol [119], nicotine [128], and psychedelics [157] alter shoal cohesion and shoal dynamics.

Shoals are hierarchically structured and fighting impacts the individuals’ behavior and hierarchical standing in the group. In social conflict situations (fighting) the dHbl-dIPN pathway gets activated in “winner” fish, whereas “loser” fish activate the dHbm-vIPN pathway (see also social defeat paradigm). Consequently, fish with silenced dHbl-dIPN pathway tend to lose fights whereas dHbm-vIPN silenced fish tend to win [62]. These findings indicate that the habenula is an important regulator of social behavior and that the different habenular nuclei have seemingly opposing roles in this process.

Our overview highlights the availability of numerous behavioral tests, some of which were already applied to study behaviors related to habenular linked disorders (Table 1). Admittedly, schizophrenia, depression, autism, and bipolar disorders are too complex to be disentangled to their very core by analyzing animal behaviors. Moreover, distinguishing psychiatric phenotypes, for instance, anxiety versus depression-like behavior, is generally challenging since the behavioral responses assessed are similar and sometimes even overlapping [158,159]. It also has to be kept in mind that a particular behavior in a fish, which looks like it would mirror a certain behavior in humans only by the looks of it might in fact resemble something very different. However, the availability of pharmaceutical drugs, which evidently have effects in zebrafish related to those seen in mammalian systems, raises hopes for future generations of standardized symptomatic readouts and treatments also independently of a precise categorization of the underlying mental disorder.

## 4. Critical Review and Outlook

Mental illnesses have a high hereditary component and various potentially causative risk genes have been uncovered. The versatile zebrafish model is excellently suited to link such risk genes to neural circuits like the habenular network and help to decipher disease mechanisms [24,25]. Perhaps surprisingly, this network in the zebrafish brain has not been reported as a model of complex psychiatric disorders to date despite its great potential that we have summarized here. Experimental set-ups to study behavioral responses in habenular related disorders like anxiety, depression, and social behavior are established (Table 1). Importantly, pharmaceutical drugs used in patients appear to similarly ameliorate behaviors also in the fish model. Together with the availability of various tools and resources to insert and control for defined manipulations of the habenular neural network to study the functional consequences in vivo, this system appears very powerful. An additional invaluable method applicable in the transparent zebrafish embryo in this context is functional imaging of complex brain-wide network activities in vivo, which has been reviewed recently (i.e., [160]). At this point, a robust standardization of the tests and readouts is needed to move to the next level and use the advantages of the fish for in vivo large-scale screenings to uncover disorder ameliorating compounds. These could be tested subsequently in the respective mammalian disease models. Efforts to catalog and define zebrafish behavioral phenotypes have been undertaken (zebrafish behavior catalog, ZBC) for both larval and adult models [113]. However, consistency of the observed phenotypes is sometimes lacking between different laboratories (e.g., light/dark preference). One argument, which is often put forward to explain this caveat, is the use of different wildtype strains, which show a broad variety of responses to stress, different baselines for stress/anxiety, and different outward cues [161]. Naturally, different wild-type strains have non-identical genetic backgrounds. But even within the wild-type strains, the genomes of individuals exhibit great variabilities. Inbreeding of fish for many generations to counter this variability is currently very difficult, which makes reproducibility and hence the standardization of the different tests time-consuming. Another aspect to consider when working with zebrafish behavior is that addressing questions regarding gender-related differences is not trivial. Gender frequently plays a role in behaviors and also the treatment of specific disorders [162,163]. Sex determination in zebrafish is not well understood and in larvae, with which many behavioral tests are carried out and at young adolescent stages sex of zebrafish cannot be determined. Therefore, no conclusions as to the influence of sex differences can be drawn before fish reach adulthood. Considering these issues, one should keep other well-established fish model systems in mind such as the medakafish (*Oryzias latipes*). Medaka wild-type strains are highly inbred and sex determination genes are well studied [164,165]. Although the medaka research community is growing, resources and tools for behavioral studies and habenular development and function are however still rather limited [58,166,167].

As with all animal models of human disorders, we should be careful to anthropomorphize the animal. Animal behaviors and responses are not as complex as human behaviors and certain drivers of diseases like society and lifestyle cannot easily be modeled. With respect to the habenular network, the main connections and nuclei are conserved throughout vertebrates. However, the system is far less complex in fish compared to humans. This is a great advantage for addressing neuroscientific questions by studying the development and function of an entire network as a whole. But we also need to carefully consider this aspect when interpreting our data. The strength of zebrafish as a model is not necessarily the precise modeling of human behaviors in their entire complexity. The power of the system lies in discovering and dissecting conserved genetic and physiological mechanisms underlying mental disorders in vivo and the study of common core behaviors. Now the tools and resources are available to rapidly identify new therapeutic targets and approaches to ameliorate habenular-related disorders using the great possibilities the zebrafish system offers.

## Figures and Tables

**Figure 1 biomolecules-11-00324-f001:**
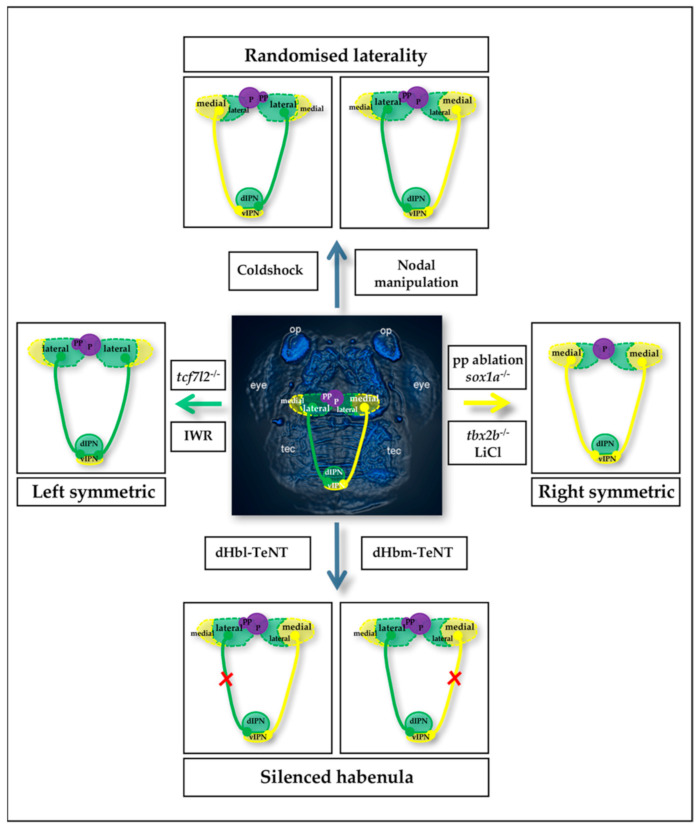
Overview of zebrafish habenular network manipulations. Dorsal view on the head of a 4-day old embryo labelled for anti-acetylated tubulin, anterior to the top (central picture). Highlighted are the pineal complex and the dorsal habenula including their efferent projections to the interpeduncular nucleus (IPN) only. Efferent axons from the lateral (green) and the medial (yellow) habenular neurons project to the dorsal and ventral IPN, respectively. Manipulation strategies to change laterality of asymmetry of the habenulae are shown on the top and those causing symmetry on the left and right. For silencing dHbl neuron-IPN transmission, Tg(narp:GALVP16); Tg(UAS:TeNT); Tg(brn3a-hsp70:GFP) (dHbl neurons) transgenic lines are used, while Tg(gpr151:GAL4VP16); Tg(brn3a-hsp70:GFP-Cre); Tg(UAS:loxP-DsRed-loxP-GFP-TeNT) are used for silencing of the dHbm-IPN information flow (lower pictures [62]). op, olfactory placode; P, Pineal; pp, parapineal; tec, tectum.

**Table 1 biomolecules-11-00324-t001:** Behavioral tests to study habenula linked disorders.

Human Disorder	Zebrafish Test	Observed Behavior/Phenotype	Stage	Rodent Test	Refs
Anxiety disorder	Novel tank test	Diving and reduced exploration behavior, thigmotaxis, erratic swimming and freezing.	Adult	Open field test, elevated plus maze	[76,77,78,79,80]
Anxiety disorder	Open field test	Thigmotaxis, hyperactive swimming, erratic swimming, freezing	Adult	Open field test	[81,82,83]
Anxiety disorder	Aversive cue exposure (shadow, predator, alarm substance)	Erratic movements, freezing post aversive cue	Adult	Predator, predator image, odor exposure	[76,84,85,86,87,88]
Anxiety disorder	Light–dark box	Avoidance of bright area in adults (scototaxis) and dark area in larval fish	Adult, larva	Light–dark box	[89,90,91,92]
Anxiety disorder	Sudden light-to-darkness transition	Increased thigmotaxis	Larva		[93]
Anxiety disorder	Electric shock assay	Increased freezing bouts post single, unconditioned shock	Larva		[94]
Anxiety disorder	Cued fear-conditioning tasks	Reduced flight behavior to conditioned aversive stimulus	Adult, larva	Cued fear-conditioning tasks	[53,95]
Anxiety disorder, substance abuse, withdrawal	Shoaling test	Increased shoal density, smaller distance between individual fish, scototaxis, thigmotaxis	Adult	Social behavior test	[96,97,98]
Anxiety disorder, schizophrenia	Startle test	Reduced escape response to unexpected and/or aversive cues, impaired prepulse inhibition (PPI)	Adult, larva	Startle test	[99,100,101,102,103]
Depression like state	Repeated social defeat in fights	Loss of motivation to fight and win	Adult	Social defeat model	[104]
Depression like state	Unpredictable chronic stress (UCS)	Elevated anxiety levels, disturbed social interaction	Adult, larva	Unpredictable chronic mild stress (UCMS)	[105,106,107]
Depression like state	Prolonged unpredictable strong chronic stress (PUCS)	Anxiety-like and motor retardation-like behaviors	Adult	Unpredictable chronic stress (UCS)	[108]
Depression like state	Behavioral challenge (BC) assays, (e.g., inescapable aversive cue)	Initial escape response followed by reduced overall locomotive activity	Larva	Inescapable shock	[109]

**Table 2 biomolecules-11-00324-t002:** Pharmacological approaches to study habenula linked disorders.

Human Disorder	Compound	Observed Behavior	Stages	Refs
Anxiety disorder, depression	GABA-ergic: ethanol, chlordiazepoxide, diazepam	Anxiolytic in anxiety paradigms	Adult, larva	[77,79,82,83,116,119,121,122,123,124,125]
	Serotonergic: buspirone, citalopram, desipramine, fluoxetine, ketamine, LSD, olanzapine			
	Cholinergic: nicotine, scopolamine			
	Histaminergic: α-fluoromethylhistidine			
	Glutamatergic: MK-801			
	Adenosinergic: caffeine	Anxiogenic in anxiety paradigms		
	GABA-ergic: FG-7142			
Substance abuse and withdrawal	Alcohol, benzodiazepines, barbiturates, opioids (e.g., cocaine, morphine), psychostimulants (e.g., LSD), nicotine	Erratic behavior, freezing, reduced diving and explorative behavior, shoaling, increased aggressive behavior	Adult, larva	[70,126,127,128]
Substance abuse, Fetal alcohol syndrome (FAS)	Alcohol	Hyperlocomotion at lower doses and hypolocomotion at higher doses, longtime changes in social behavior in FAS model	Adult, larva	[129,130,131]

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
