# Peer review of "Zebrafish Tools for Deciphering Habenular Network-Linked Mental Disorders"

_biomolecules, 2021, doi:10.3390/biom11020324_

Round 1

Reviewer 1 Report

This is a well written and timely review.  The style is readable and the authors provide a wide range of references.  I have only a few suggestions for improvement.  The first suggestion is perhaps inspired by the final paragraph of the manuscript.  There is a danger of circular logic in these studies.  We need the models because we do not understand what is happening in humans: we validate our models because they seem like the processes in humans that we do not understand.  In other words, we assume that the processes in fish that resemble human anxiety or depression are in fact equivalent to the human state.  The common ancestor of fish and humans lived a very long time ago.  Did they really exhibit depression?  I suggest that the authors add more to address this potential weakness in the model.  More criticality would strengthen rather than weaken the model.

Another related idea I think might be added is training of the investigator.  I personally was introduced to fish keeping in grade school, long before I entered a laboratory.  This provided a background that greatly helped my research with zebrafish.  The authors might caution investigators new to the field that while simple and inexpensive to keep, zebrafish behavior is not a turn key system.  Some knowledge of fish natural history is important for consistent results.

Author Response

Reviewer 1: This is a well written and timely review.  The style is readable and the authors provide a wide range of references.  I have only a few suggestions for improvement.  The first suggestion is perhaps inspired by the final paragraph of the manuscript.  There is a danger of circular logic in these studies.  We need the models because we do not understand what is happening in humans: we validate our models because they seem like the processes in humans that we do not understand.  In other words, we assume that the processes in fish that resemble human anxiety or depression are in fact equivalent to the human state.  The common ancestor of fish and humans lived a very long time ago.  Did they really exhibit depression?  I suggest that the authors add more to address this potential weakness in the model.  More criticality would strengthen rather than weaken the model.

Response: Thank you very much for your positive evaluation. We perfectly agree that zebrafish (and animal models in general) cannot recapitulate the full spectrum of human behaviors and we have to be very cautious to over-interpret any results we obtain. We touched upon this issue at the ends of sections 3.2 and 3.3. as well as section 4. In addition, we now adjusted the abstract (page 1, line 24): “We further give an overview of existing behavior assays to address phenotypes related to mental disorders and critically discuss the power but also the limits of the zebrafish model for identifying suitable targets to develop therapies.” We also changed a part at the end of the introduction (page 2, lines 59-60): “Many of these behaviors can be studied in animal models including the zebrafish. Even if the full complexity of human behaviors cannot be mimicked in animals, the modeling of some typical aspects might serve as an excellent readout for identifying disorder related risk genes.” Finally, we inserted a sentence at the end of section 3.3 (page 5, 6, lines 216-219): “It also has to be kept in mind that a particular behavior in a fish, which looks like it would mirror a certain behavior in humans only by the looks of it might in fact resemble something very different.”

Reviewer 1: Another related idea I think might be added is training of the investigator. I personally was introduced to fish keeping in grade school, long before I entered a laboratory. This provided a background that greatly helped my research with zebrafish.  The authors might caution investigators new to the field that while simple and inexpensive to keep, zebrafish behavior is not a turn key system.  Some knowledge of fish natural history is important for consistent results.

Response: We perfectly agree with the reviewer. In fact, we found the expression “turnkey system” very much appropriate and allowed ourselves to “steal” it. We inserted a sentence at the end of the introduction to section 3 (page 12, lines 384-386): “It is important to emphasize at this point that zebrafish behavior is not a turnkey system and obtaining consistent results requires meticulous preparation, profound knowledge and time for setting up a robust test system (see also the concluding section 4).”

Reviewer 2 Report

The manuscript entitled „Zebrafish tools for deciphering habenular network linked mental disorders” by Bühler and Carl represents a comprehensive summary of findings underlying the hypothesis that the habenular network and its role at the centre of communication from and to different vertebrate brain regions is important in the development of neuropsychiatric disorders. As the habenular network is present in vertebrate model organism as well, this system can be investigated using different techniques and organisms. The authors highlight in the second part the habenular system in the model organism Danio rerio and describe its features and relation to the mammalian systems. Furthermore, they present several transgenic lines which uncovered defects in habenular plasticity and describe its developmental dependence on signalling systems such as Wnt/beta-catenin/notch and transcription factor dependence like Sox1a and tbx2b. To me, the last and third part of this review is the most important for the scientific community. The authors decided to present well-known zebrafish behavioural assays from a different perspective. First, they focus of the habenular contribution in the human condition and then describe behavioural assays to investigate this in the fish. They focus on anxiety, depression, substance abuse and social interaction. The author concludes with an outlook how danio rerio can be exploited more as a model system for human psychiatric conditions if researchers are interested in the contribution of the habenular system in these pathologies.

The review is well written, the decisions of the authors which behavioural assays to present and focus on certain psychiatric conditions is absolutely justified. The conclusions from the review are discussed appropriately.

There are no flaws associated with this review. The literature presented seems to me appropriate given the focus the author decided to deliver here. Therefore, I would recommend the publication of this review to inform more studies with habenular contribution in psychiatric conditions and the use of Danio rerio as a valuable model organism. Minor points which could be addressed before publication:

  1. Page 2 (line 46): “depressionby” should read “depression by”.
  2. Page 2 (line 76): there is a strange and useless bracket after the citation.
  3. Page 4 (Figure 1): I like this layout and the staining. But maybe it would be easier for the reader to grasp the concept if this would be presented as a schematic drawing rather than showing as stainings to highlight the major points of habenular manipulations. Just a didactic suggestion.
  4. Page 6 (Table 1): I was wondering if it would make sense to group the assays in larvae and adult assays? Could help the reader to decide whether certain aspects can be assessed easily in their preferred developmental stage.
  5. Page 9 (Table 2): Very useful table. Could be improved if a more detailed description is given (for instance: GABAergic compounds (which?), influence (positively or negatively) this and that behaviour, and so on. Could help researchers chose more carefully specific compounds for their experiments and research questions).

Author Response

Reviewer2:

1.) Page 2 (line 46): “depressionby” should read “depression by”.

2.) Page 2 (line 76): there is a strange and useless bracket after the citation

Response: Thank you very much for your positive evaluation. Both typing errors on page 2 (line 46 and 78) have been corrected.

Reviewer2: Page 4 (Figure 1): I like this layout and the staining. But maybe it would be easier for the reader to grasp the concept if this would be presented as a schematic drawing rather than showing as stainings to highlight the major points of habenular manipulations. Just a didactic suggestion.

Response: We agree that Figure 1 can be improved to make it easier to understand. We left the central picture underlying the schematic for orientation within the brain and removed the background along with your suggestion. We also improved the general presentation.

Reviewer2: Page 6 (Table 1): I was wondering if it would make sense to group the assays in larvae and adult assays? Could help the reader to decide whether certain aspects can be assessed easily in their preferred developmental stage.

Response: Thank you for the suggestion, but as the emphasis of the review is on human disorders, we decided to group the assays according to the disorder they are used to investigate. In Table 1 (page 6), column “stage”, we specify which assays can be used in adults or larval stages or both. Grouping the assays into larvae and adult assays would lead to multiple nominations of the same assay, which we believe is not more reader-friendly.

Reviewer2: Page 9 (Table 2): Very useful table. Could be improved if a more detailed description is given (for instance: GABAergic compounds (which?), influence (positively or negatively) this and that behaviour, and so on. Could help researchers chose more carefully specific compounds for their experiments and research questions).

Response: Thank you for your positive feedback and excellent suggestion. We agree that more details on the compounds already tested in each group might be of interest to the readers. Therefore, we specified the compounds and grouped them in anxiolytic and angiogenic compounds in Table 2 (page 9). A detailed description of anxiolytic and angiogenic effects can be found in section 3.1 in the main text.